# Genome-wide identification of genes involved in beetle odoriferous defensive stink gland function recognizes Laccase2 as the phenoloxidase responsible for toxic *para*-benzoquinone synthesis

**Bibi Atika[1]**ⓔ, **Sabrina Lehmann[1]**ⓔ, **Elisa Buchberger[1]**, **Musa Dan'azumi Isah**ⓘ[1], **Zahra Basirkazerouni[1]**, **Michael Rostás[2]**, **Gregor Bucher**ⓘ[3], **Ernst A. Wimmer**ⓘ[1]*

1 Department of Developmental Biology, GZMB, Johann-Friedrich-Blumenbach-Institute for Zoology and Anthropology, Georg-August-University Goettingen, Goettingen, Germany, 2 Agricultural Entomology, Department of Crop Sciences, Georg-August-University Goettingen, Goettingen, Germany, 3 Department of Evolutionary Developmental Genetics, GZMB, Johann-Friedrich-Blumenbach-Institute for Zoology and Anthropology, Georg-August-University Goettingen, Goettingen, Germany

ⓔ These authors contributed equally to this work.
* ewimmer@gwdg.de

## Abstract

Exocrine glands have evolved several times independently in Coleoptera to produce defensive chemical compounds with repellent, antimicrobial, or toxic effects. Research on such glands had focused on morphological or chemical ecology methods. However, modern genetic approaches were missing to better understand this biological process. With the rise of the red flour beetle, *Tribolium castaneum*, as a model for studies of development and pest biology, molecular genetic tools are now available to also study the safe generation of toxic compounds in defensive stink glands. Using the RNA-interference-based, genome-wide, phenotypic screen "iBeetle" and the re-analysis of gland-specific transcriptomics based on a significantly improved genome annotation, we could identify 490 genes being involved in odoriferous stink gland function. In the iBeetle screen, 247 genes were identified, of which we present here 178 genes identified during iBeetle's 3rd phase, while the transcriptomics analyses identified 249 genes, with six genes being identified in both functional genomics approaches. Of these 490 genes, only about 40% of these genes have molecularly characterized homologs in the vinegar fly, while for 213 genes no fly homologs were recognized and for 13 genes no gene ontology at all was identified. This highlights the importance of genome-wide gene identification in tissues that have not been previously analyzed to recognize potentially new gene functions. Gene ontology analysis revealed "SNARE interactions in vesicular transport", "Lysosome", "Pancreatic secretion", and "MAPK signaling pathway – fly" as key pathways. Additionally, many of the genes are encoding enzymes, transcription factors, transporters,

**Data availability statement:** The datasets of the iBeetle screen including the fragments used as double-strand RNAs for the gene knock-downs are available in iBeetle-Base (https://ibeetle-base.uni-goettingen.de/) https://doi.org/10.1093/nar/gku1054, https://doi.org/10.1093/nar/gkx984. All other relevant data are in the manuscript and its supporting information files.

**Funding:** 2. The first two phases of the iBeetle screen were carried out as central projects of DFG Research Unit "iBeetle" (FOR 1234), which was funded by the German Research Foundation (G.B.: BU 1443/7-1, BU 1443/7-2). The continuation into the third phase was funded by Bayer CropScience (G.B.). B.A. was supported by the Erasmus Mundus Action 2 project SmartLink: South-East-West Mobility. We acknowledge support by the Open Access Publication Funds of the Göttingen University (E.A.W.). The funders had no role in study design, data collection and analysis, decision to publish, or preparation of the manuscript.

**Competing interests:** The authors have declared that no competing interests exist.

**Abbreviations:** 1-C15, 1-pentadecene; 1-C17, 1-heptadecene; dsRNA, double-stranded RNA; EBQ, 2-ethyl-1,4-benzoquinone; FC, Fold change; GC-MS, gas chromatography-mass spectrometry; MBQ, 2-methyl-1,4-benzoquinone; OGS, official gene set; RNAi, RNA interference; RNA-seq, next generation sequencing of mRNA; Tcas5.2, official assembly of genomic sequence of *Tribolium castaneum* version 5.2.

or are involved in membrane trafficking. As the phenoloxidase responsible for generating the toxic *para*-benzoquinones in the stink glands of the beetle, we could identify *laccase2*, which is expressed in the last secretory cell in contact with the cuticle-lined vesicular organelle, where the toxic compounds are safely produced before being released into the gland reservoir.

## Author summary

Certain beetles produce toxic substances to defend themselves against microorganisms or predators. For the generation of such chemicals, the beetles have specialized glands, which are organized in a way that they can produce and store the toxic compounds without harming themselves. The morphology and biochemistry of such glands had been intensively studied. However, the genes that are involved in this process had not been recognized. Here we present the identification of about 500 genes that are involved in the function of beetle defensive glands, which presents a starting point for a detailed molecular understanding of the process of safe production of toxic substances. A key feature for the production is the use of non-harmful precursors that are only finally processed by a specifically secreted enzyme called Laccase2 to become toxic in a cuticle-shielded organelle.

## Introduction

Exocrine glands are widespread in the animal kingdom and represent specialized secretory structures that release chemical substances to the external surface of the body or into a reservoir for storage [1,2]. The functions of their secretions fall into three general categories: i) physiology, e.g., waterproofing of the exoskeleton, improving digestion, or producing fibrous secretions for protective structures; ii) communication, such as pheromonal signaling; and iii) defense to protect from attack by spreading repellents or toxins [1]. In the Coleoptera, defensive glands seem to have arisen many times independently and are usually multifunctional by producing chemical compounds that operate as repellents, surfactants, antimicrobics, or toxicants against a large array of potential target organisms [3,4]. In many beetles, the defensive compounds are produced in complex glands, stored in cuticle-lined extracellular reservoirs, and released by controlled opening [5]. To secrete defensive compounds, the exocrine gland must acquire secretions, translocate them across the cell membrane to extracellular compartments, store them, and release them in a regulated manner [2]. For the acquisition of secretions, substances must be de-novo synthesized or sequestered by the uptake of dietary precursors [6,7], which need to be transported to and taken up by the glands [8,9], where the final secreted compounds are produced. The translocation to the extracellular compartment can either happen by passive diffusion of small apolar organic molecules [1] or by specific transporters

[9]. The storage of larger amounts of defensive secretions necessitates a cuticle-lined reservoir to protect the gland cells especially when toxic substances are produced [10]. When the secretion of compounds from reservoirs is controlled, often specific muscles are involved that dilate the gland opening [11].

Insect exocrine glands have been researched to a great amount. However, most studies so far focused on structural approaches to describe the morphology or on chemical ecology methods to identify the secretions [2]. That modern genetics should make more impact on the field of semiochemical and defensive compound secretion was pointed out already thirty years ago [12]. However, so far only very few molecular genetic studies have focused on defensive gland function. The major genetic insect model, *Drosophila melanogaster*, does not have such glands, which is probably the reason for the limited amount of studies in this field. The chemical defense system of rove beetles, the tergal gland, has recently become a key example for understanding cell type evolution underlying biosynthetic innovation [13,14]. Within the coleoptera, however, the red flour beetle, *Tribolium castaneum*, represents the most powerful genetic model organism with an elaborate genetic tool kit [15] and unbiased large-scale RNA interference (RNAi) screening to study biological processes [16]. *Tribolium* beetles (Coleoptera: Tenebrionide) possess paired odoriferous defensive stink glands in the prothorax (thoracic glands) and the posterior abdomen (abdominal glands) [10]. They are composed of two types of secretory units (cell type 1 and cell type 2) that harbor cuticle-lined so-called vesicular organelles, which are connected with tubules to a large reservoir that can be opened by respective muscles [10,11,17]. The glands are filled with large amounts of highly reactive, unstable, and toxic *para*-benzoquinone compound [5,10,17–21]. In addition, rare 1-alkenes – called terminal olefins – are found [22–26]. Self-protection against the toxic substances is conferred by producing the *para*-benzoquinone compounds only in the cuticle-lined vesicular organelles [17]. In *T. castaneum*, cuticle sclerotization – hardening based on molecular cross-linking under enzymatic control – requires approximately four days for completion [27], which probably is the reason why newly emerged *Tribolium* adults lack defensive secretions [21,28]. The beetles use the *para*-benzoquinone compounds to condition their microenvironment against microorganismal growth [21], which turns the flour also unusable for baking [29] and hazardous to human health [30].

To identify large sets of genes that are involved in a specific biological process or the function of a particular tissue, functional genomics approaches such as unbiased systematic genome-wide gene disruption screens or comparative transcriptomics analyses of gene expression profiles from different tissues are of particular importance [31]. In this respect, the start of a genome-wide RNAi screen in *T. castaneum* – termed iBeetle – based on systematic gene knock-down and broad phenotypic analysis first revealed 32 genes affecting gland function when 5,300 genes had been analyzed (1st phase) [32], which could be increased to 69 genes causing phenotypic changes in glands when 8,500 genes had been screened (2nd phase) [33]. In continuation and finalization of the screen (3rd phase), now 13,020 genes representing 78% of the 16,593 currently annotated genes [34] have been analysed [35]. Here, we present the 178 additional genes affecting gland function identified out of the 4,520 genes screened in the 3rd phase. During the iBeetle screen, all observed phenotypic changes caused by the RNAi-mediated knock-down of the genes were systematically and methodically documented in iBeetle-Base [36,37]. In addition, a first transcriptomics analysis identified 77 gland-specifically expressed genes in *T. castaneum* in 2013 [38] based on the official gene set 2 (OGS2) [39]. In 2020, a new genome assembly (Tcas5.2) and enhanced genome annotation for *T. castaneum* resulted in a new official gene set (OGS3), which led to the discovery of 1452 novel gene sequences [34]. Because of the improved genome annotation, we re-analyzed the original transcriptomics data [38] with the new OGS3 and present here 249 genes that are clearly higher expressed in gland tissue compared to a control tissue, the anterior abdomen.

Since the hypothesized model for *para*-benzoquinone synthesis proposed the involvement of beta-glucosidases, phenoloxidases, and peroxidases [17], Li et al. [38] analyzed all such annotated enzymes for their relative gland transcriptome expression level. Out of the phenoloxidases, the gene coding for Laccase2 was clearly gland upregulated. However, because of the high cut off rate of being 64x higher expressed in the gland tissue compared to an abdominal control tissue, the gene was not declared gland-specific [38]. *laccase2* (Tc_010490; iB_12548) has now also been detected in

the 3rd phase of the phenotypic iBeetle screen as being involved in gland function. Therefore, we present here the novel function of *laccase2* in *para*-benzoquinone production in *T. castaneum* odoriferous stink glands. Laccase2 belongs to the family of multicopper oxidases [40] and has in *T. castaneum* previously been described to be involved the oxidation of precursors to generate *ortho*-benzoquinones used in cuticle sclerotization and pigmentation of the beetle [41,42].

Altogether, here we present the genome-wide identification of 490 genes involved in odoriferous stink gland function by transcriptomics analysis and a systematic unbiased phenotypic RNAi screen (iBeetle), as well as the stink gland function of the phenoloxidase Laccase2 in generating toxic *para*-benzoquinones.

## Results and discussion

### iBeetle: Genome-wide RNAi-based phenotypic screen

iBeetle represents a systematic phenotypic screen that applies RNAi in a genome-wide forward genetics way [32,35]. In the first two phases (1st and 2nd) of iBeetle, a total of around 8,500 genes (50% of the annotated genes) were screened, which identified 69 genes whose knock-down caused phenotypic changes in gland morphology and partially in gland secretion content [33]. In these two phases, iBeetle served as a first-pass screen to identify potentially interesting genes for a particular biological process that needed to be re-screened to verify their phenotype, since nine independent screeners checked simultaneously for many embryonic and postembryonic phenotypes without systematically dissecting out the glands for detailed inspection. Thus, in this first pass screen most defensive stink gland phenotypes were identified in intact adult beetles through the cuticle, which made it very likely that a considerable amount of genes were missed. Only in the re-screen of the genes annotated as gland-affecting, systematic dissections of the glands were performed [33].

Here, we present the result of the 3rd phase of the iBeetle screen covering an additional 4,520 genes, in which the analysis for phenotypes affecting the defensive stink gland was carried out directly by dissection and detailed inspection of the gland tissue to better cover the detection of phenotypes. In this part of the screen, we could identify 178 genes whose knock-down affects gland morphology and partially also the gland volatile composition (S1 Table). However, due to the number of identified genes, we have so far not carried out a second analyses with non-overlapping fragments to confirm the phenotypes independently of the iBeetle-Base documented fragments [36,37] used during the screen. The detection of defensive stink phenotypes has been also documented in iBeetle-Base, which thus serves as a database for both sequence as well as phenotype information regarding odoriferous stink gland genes of *T. castaneum*.

The phenotypic analysis of the dissected stink glands revealed morphological changes that were categorized into seven groups similar to our previous study [33]: secretion color darker (I), secretion color lighter (II), irregular reservoir size (III), less secretion (IV), colorless secretion (V), melanized gland content (VI), and size decreased (VII). Examples for the different phenotypic categories are depicted in Fig 1A-1H. For the 178 genes identified in the 3rd phase of the iBeetle screen, the RNAi-based knock-down caused in most cases less secretion (44%), size-decreased glands (25%), or a darker color of the secretions (12%). The other phenotypes were only observed in few cases (Fig 1I and S1 Table).

When morphologically altered phenotypes of the glands were identified in the screen, glands from three ten-day old RNAi-treated adults were analysed by semi-quantitative gas chromatography-mass spectrometry (GC-MS) independently for the thoracic and abdominal glands to detect the main gland volatiles of *T. castaneum* [26], which are similar in males and females [38]: the *para*-benzoquinones 2-Methyl-1,4-benzoquinone (MBQ) and 2-Ethyl-1,4-benzoquinone (EBQ), as well as the alkenes 1-Pentadecene (1-C15) and 1-Heptadecene (1-C17). For 144 of the 178 morphologically identified genes, also phenotypic changes in the gland volatiles could be detected (S1 Table and Fig 2A). For 93 genes, the RNAi-based knock-down caused a strong reduction of the *para*-benzoquinones in both the abdominal and thoracic glands, with forty genes causing a reduction of all detectable volatiles, whereas 49 genes caused a gland-specific reduction of volatile compounds (Fig 2A and S1 Table). Typically, the two alkenes or the two *para*-benzoquinones were respectively affected together, while the two compound groups behave rather independently. However, there is no clear correlation between morphology and the gland volatile phenotype, which indicates that very different causes can result in the lack of volatile

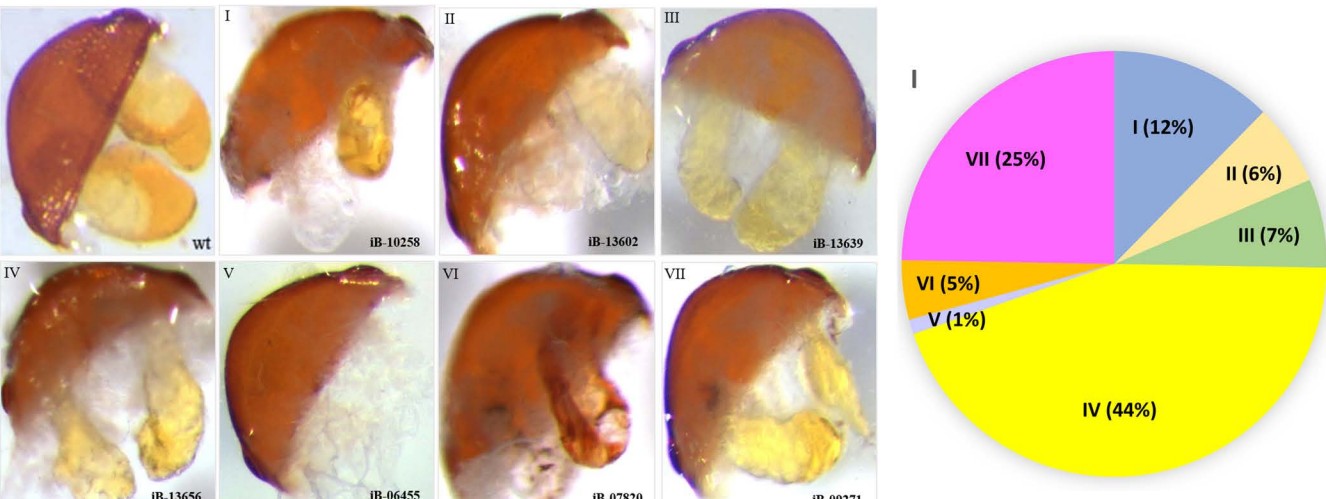

**Fig 1. Visible morphological stink gland phenotypes identified in specific gene knock-downs of the 3rd phase of iBeetle.** Morphologies differing from wild type (**A**) were categorized into seven groups: (**B**) secretion color darker (**I**), (**C**) secretion color lighter (**II**), (**D**) irregular reservoir size (**III**), (**E**) less secretion (**IV**), (**F**) colorless secretion (**V**), (**G**) melanized gland content (**VI**), and (**H**) size decreased (**VII**). To not disrupt the glands and for size comparison, the last abdominal segment, which has a width of 0.7–0.8 mm, is depicted along with the glands. The pictures for the representative phenotypes were taken from knock downs of the genes indicated by the respective iBeetle number. For the 178 genes identified, the different morphologies were recognized by the indicated percentages in panel **I** (S1 Table).

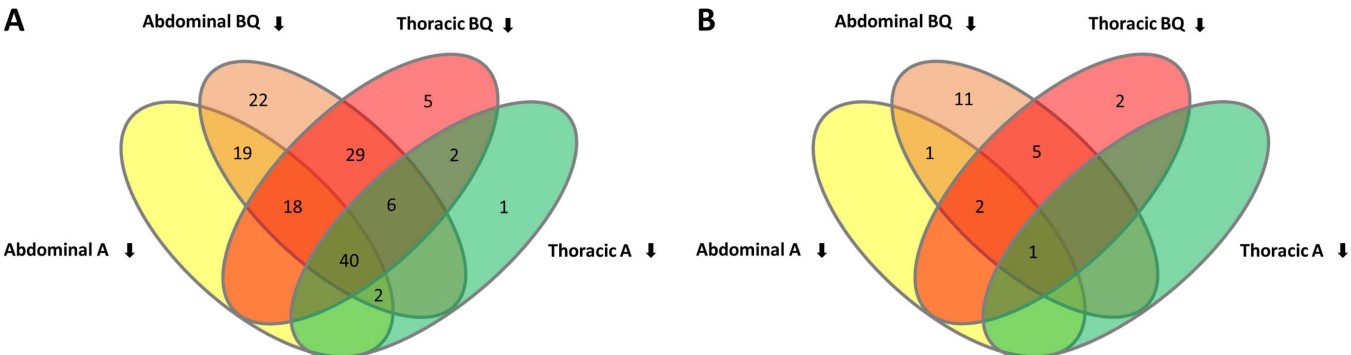

**Fig 2. Changes in stink gland volatile compound content in knock-down beetles, of (A) 178 identified genes by the 3rd phase of iBeetle (S1 Table) and (B) 41 genes revealed by our transcriptomic re-analysis that had not been identified previously (S5 Table).** Reduction was identified for the indicated number of genes in benzoquinones (BQ) or alkenes (A) in abdominal or thoracic stink glands. BQ represents benzoquinones 2-Methyl-1,4-benzoquinone (MBQ) and 2-Ethyl-1,4-benzoquinone (EBQ), while A represents the two alkenes 1-Pentadecene (1-C15) and 1-Heptadecene (1-C17).

compounds. Moreover, it should be noted that knock-down effects on volatile compound composition will have to be confirmed by additional experiments, when single gene analyses are carried out, since measurements during the screen have been done only once.

While in the first two phases of the iBeetle screen only less than 1% of the analyzed genes (69 out of 8500) were identified to be involved in stink gland function, this increased in the 3rd phase to about 4% (178 out of 4520). One reason for this greater efficiency is probably the direct analysis of the dissected glands during the screening process, since many of the genes were identified by phenotypes such as less secretion (44%) or size-decreased glands (25%), which would

have been hard to detect without dissection through the adult cuticle as performed in the first two phases of the screen [33]. Another reason, however, could also lie in the kind of genes that were analyzed during the 3rd phase of iBeetle. First gene annotations of new genomes often rely on identification of homologous genes and thus often do not represent the full repertoire of the species gene content. At the start of iBeetle, the targeting of genes was based on the OGS2 [39], while at the time of the 3rd phase of iBeetle, the *T. castaneum* genome was re-annotated and substantially improved by the inclusion of transcriptomics data including also gland transcriptomes [34], which allowed for the annotation of also more species-specific genes. Since no other well annotated genome of an arthropod having defensive stink glands had been established before *T. castaneum*, many stink gland-specific genes probably have only been annotated in the enhanced genome assembly providing OGS3 [34] and thus could only be screened in the 3rd phase of iBeetle.

**New gene set-based re-analysis of stink gland transcriptome**

To evaluate, whether and to which extent the genes identified in the 3rd phase of iBeetle are expressed in stink gland tissue, we needed to re-map the available transcriptomic data [38] to the enhanced genome annotation [34]. For this re-analysis, we used the published RNA-seq data of female abdominal glands, female thoracic glands, male abdominal glands, and male thoracic glands and as control sample the anterior abdomen of wild type *T. castaneum* beetles [38]. By using the OGS3 of the enhanced genome [34], the mapping rate could be increased by almost 30%, from 49% - 61% in Li et al. [38] to 79% - 86% in our new analysis, which is most probably due to the increased quality of the newly annotated gene set. In Li et al. [38] the identification of the 77 genes to be functionally analyzed was based on a 64x higher expression in gland tissue compared to the control tissue. However, altogether 212 genes were identified as potentially gland-specifically expressed and divided into ten different substraction subgroups with groups 1, 2, 3, 5, 9, and 10 containing genes [38]. Groups 4 and 6 did not result in gene identification and group 7 (male-specific abdominal gland) had with 299 genes probably mostly false candidates because of a likely contamination with male accessory gland tissue. The originally identified 212 genes are now presented by 208 gene models, because of gene model fusions in the enhanced genome annotation [34]. We used these 208 genes for our comparisons as the genes identified by Li et al. [38] (OGS2: Groups 1, 2, 3, 5, 9, 10; S1 and S3 Figs).

To declare genes stink gland-specifically expressed in our transcriptome re-analysis, the genes had to be at least 16x higher (log2 FC ≥ 4) expressed in all four gland tissues (female abdominal glands, female thoracic glands, male abdominal glands, and male thoracic glands) compared to the control tissue of anterior abdomen. Using the enhanced OGS3 (S2A Table), we could identify this way 75 genes being clearly upregulated in gland tissue (S1 Fig). For further comparison, we re-analyzed the OGS2 the same way (S2A Table), which identified 60 genes, of which only 50 genes overlapped with the OGS3 identified genes (S1 Fig). Of the Li et al. [38] identified 208 genes, 164 were not detected by our new criteria, seven were also detected by the OGS2 analysis and 37 genes were identified by all three transcriptomic analysis (depicted in green in S3 Table and S1 Fig). Altogether, the three transcriptomics analyses identified 249 potential stink gland-specifically expressed genes that were further considered in our analysis (S3 Table).

Only one of these genes was previously recognized also in the first two phases of iBeetle (Tc_005389; iB_09413), but was actually identified in all three transcriptomics analyses. Of the new 178 phenotype-identified genes in the 3rd phase, five additional genes are also detected by transcriptomic analyses genes (S4 Table and S3 Fig): Tc_008780 has also been identified by all three transcriptomics analyses; whereas Tc_015339, Tc_033142, and Tc_034897 were only detected by Li et al. [38] in groups 3, 2, and 10, respectively; and Tc_033013 is only found in the new transcriptomics analysis using OGS3. However, only 18 additional genes of the 178 3rd phase iBeetle identified ones have a more than two times higher expression in the stink gland tissue (S4 Table), which indicates that the phenotypic screen can identify genes that might not be specifically active in the stink glands but also in other tissues, but are still necessary for the function of the glands. One such example was already given with the identification of Tc_030914 encoding a copper-transporting ATPase-I (ATP7) that is neither up nor down regulated in the stink gland tissue [32]. ATP7 might actually have a common role in

secretion as indicated by its enriched expression in Malpighian tubules as indicated by the Beetle Atlas [43]. Moreover, genes involved in the generation of precursor molecules such as phenolic glucosides [17] could be active in the gut or fat body and their expression might therefore even be reduced in the gland tissue [32].

When comparing the mapped reads for the 178 genes identified in the 3rd phase of iBeetle by OGS3 versus OGS2 (S4 Table), it becomes evident, that many of the genes screened, were newly annotated genes not covered in the 2013 available gene set [38]. This confirms, that the enhanced assembly of the *T. castaneum* genome [34] allowed for the annotation of novel genes involved in stink gland function that were not annotated in the previous OGS2 [39].

Since our transcriptomic re-analysis revealed 41 genes, not being previously identified (S3 Table), we knocked-down these genes by RNAi and screened the injected beetles for morphological and volatile compound phenotypes. 19 genes caused an identifiable morphological phenotype of the categories described in Fig 1 (S5 Table), while 22 genes showed strong changes in volatile compounds of *para*-benzoquinones or alkenes in abdominal or in thoracic glands: for eight genes, the RNAi-based knock-down caused a strong reduction of the *para*-benzoquinones in both the abdominal and thoracic glands, with only one gene causing a reduction of all detectable volatiles, whereas 14 genes caused a gland-specific reduction of volatile compounds (Fig 2B and S1 Table).

The detection of tissue-specifically expressed transcripts relies heavily on the reference gene set used to map the reads against. Since genomic and transcriptomic reference genomes are constantly improving, re-mapping original RNAseq datasets, which include all transcripts that are expressed in a specific tissue, against new reference gene sets provides the opportunity to detect new genes without necessarily generating and sequencing new samples. Using the enhanced genome assembly of *T. castaneum* providing OGS3, allowed us to increase the mapping rates of the transcript reads by about 30% compared to the initial analysis and to detect 41 additional genes being comparatively highly expressed in stink gland tissue.

## Gene ontology analyses of 490 genes involved in stink gland function

While iBeetle recognized 247 genes, the transcriptomics analyses identified 249 genes, with six genes being identified in both functional genomics approaches, thus resulting in 490 genes being potentially involved in stink gland function (Fig 3). To recognize their molecular function, we first identified the closest *D. melanogaster* homologs (S6 Table) as indicated by iBeetle-Base [37]. Only for 194 genes, functionally characterized homologs could be recognized, whereas 83 genes identified uncharacterized proteins and for 213 genes not homologs were found. Finding molecularly characterized homologs in the vinegar fly only for about 40% of the genes is remarkable, however, not necessarily unexpected, since a corresponding tissue of odoriferous stink glands does not exist in the fly. Therefore, many of the identified genes might be involved in uncharacterized metabolic pathways or in self-protection against the produced and stored defensive, toxic secretions.

To potentially identify metabolic pathways those 490 genes might be involved in, we performed the gene ontology analyses ShinyGO [44], BlastKOALA [45], and eggNOG-mapper [46]. (S6 Table). As expected for a secretory tissue, most identified genes are involved in genetic information processing, signaling and cellular processes, environmental information processing, metabolism as well as in Golgi apparatus, exosome, and lysosome function (Figs 4, S2 and S3). One of the 17 genes identified for the "Lysosome" pathway, Tc_015151, encodes arylsulfatase B (ARSB), which was analysed already by Li et al. [38] with screen number GT62. The knock-down of Tc_015151 caused a complete *para*-benzoquinone-less phenotype without affecting the alkene composition causing colorless secretions in all stink glands.

Overall, only thirteen genes (four of them having *D. melanogaster* homologs of uncharacterized function) were not analyzed or did not provide results by any of the three ontology analyses. Of these genes, four had very high iBeetle numbers indicating that they were included only late in iBeetle probably due to their late gene annotation and seven do not even have an iBeetle number assigned suggesting that they were only annotated after the iBeetle screen was concluded. All those genes were identified by one or several of the transcriptomics analyses and might represent interesting candidate genes for novel molecular functions. Interestingly, Tc_033013, which was recognized by the new transcriptomics analysis

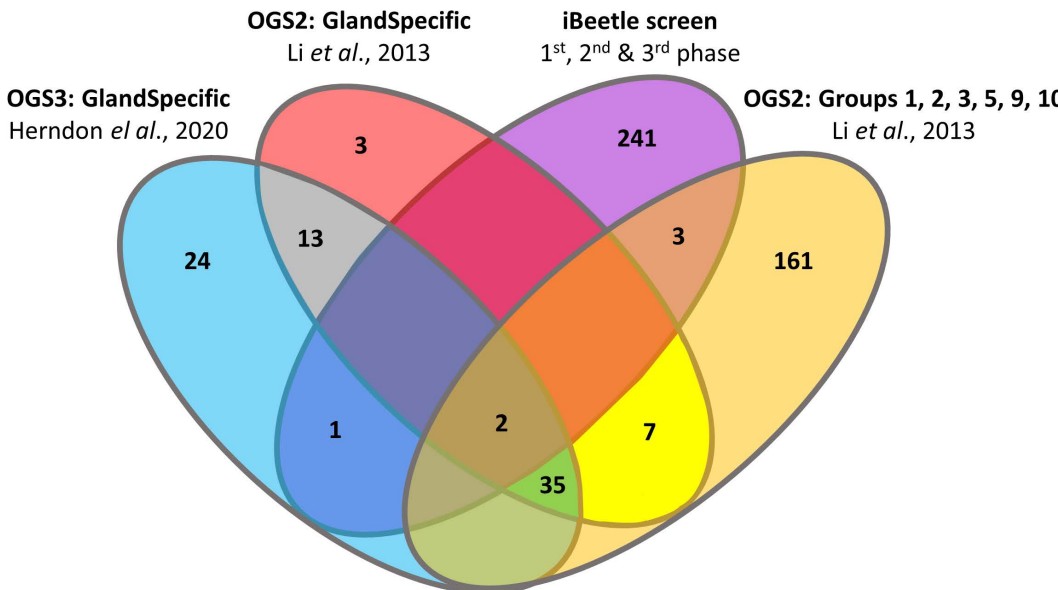

**Fig 3. Phenotypic screen iBeetle and different transcriptomics analyses recognize 490 gland genes with limited overlap.** The different transcriptomics analyses revealed together 249 genes (S1 Fig). Comparing the 247 stink gland-affecting genes identified by iBeetle (purple ellipse) with the 249 transcriptomics-identified genes, only six of them were identified by both functional genomics approaches: Tc_005389 (2nd phase iBeetle) and Tc_008780 (3rd phase iBeetle) were actually identified also in all three transcriptomics analyses; Tc_015339, Tc_033142, and Tc_034897 (all 3rd phase iBeetle) were only detected by Li et al. [38] in groups 3, 2, and 10, respectively; and Tc_033013 (3rd phase iBeetle) was only identified in the new transcriptomics analysis using OGS3 (S3 Table). Altogether the phenotypic iBeetle screen and the transcriptomics analyses thus recognized 490 genes that were subject to ontology analysis to identify their molecular function (S6 Table).

based on OGS3 and identified in the 3rd phase of iBeetle, encodes either a non-coding RNA or a hypothetical peptide of 34 amino acids (MSFVTKALLVIMVLISLTVGEVYKPPPGHHKGRK) that has no homology or gene ontology data available so far.

## Stink gland function of the phenoloxidase Laccase2 in generating toxic secretions

The hypothesized model for the synthesis of toxic *para*-benzoquinones [17] proposed the involvement of beta-glucosidases, phenoloxidases, and peroxidases. Here, we could identify the phenoloxidase Laccase2 (Tc_010490) in the 3rd phase of iBeetle. A stink gland-enriched expression of this phenoloxidase was already recognized by Li et al. [38], but it did not pass the high cut off rate of being 64x higher expressed to be considered one of the 208 gland-specific genes. Besides Tc_010490, also Tc_010489 was identified as coding for a stink gland-enriched phenoloxidase with an even more enhanced stink gland-specific expression. Both gene models were similarly not recognized as stink gland-specific in our transcriptomics analysis, with Tc_010489 almost reaching the cut off criteria, as it had log2 FC ≥ 4 for three of the four samples but in the female abdominal sample only 3,1 (S2 Table). Both Tc-numbers are still representing separate gene models in iBeetle-Base (https://ibeetle-base.uni-goettingen.de/details/TC010490), while NCBI indicates them as splice variants of *laccase2* (*Tc_lac2*), which had been recognized already before [41]. The gene structure with encoding two splice variants is actually conserved across different insect orders [47]. Tc_010489 represents the complete gene *Tc_Lac2A* including the differentially spliced exons 6a, 7a, and 8a, while Tc_010490 only represents the differentially spliced exons 6b, 7b, and 8b of *Tc_Lac2B* (Fig 5A). The overall amino acid sequence identity between the two encoded proteins amounts to 92%, whereas in the variable C-terminal region only 74% are similar [41]. Gene model Tc_010489, which is actually covered by 500x times more stink gland reads than Tc_010490 (S2 Table), was analysed in the 1st

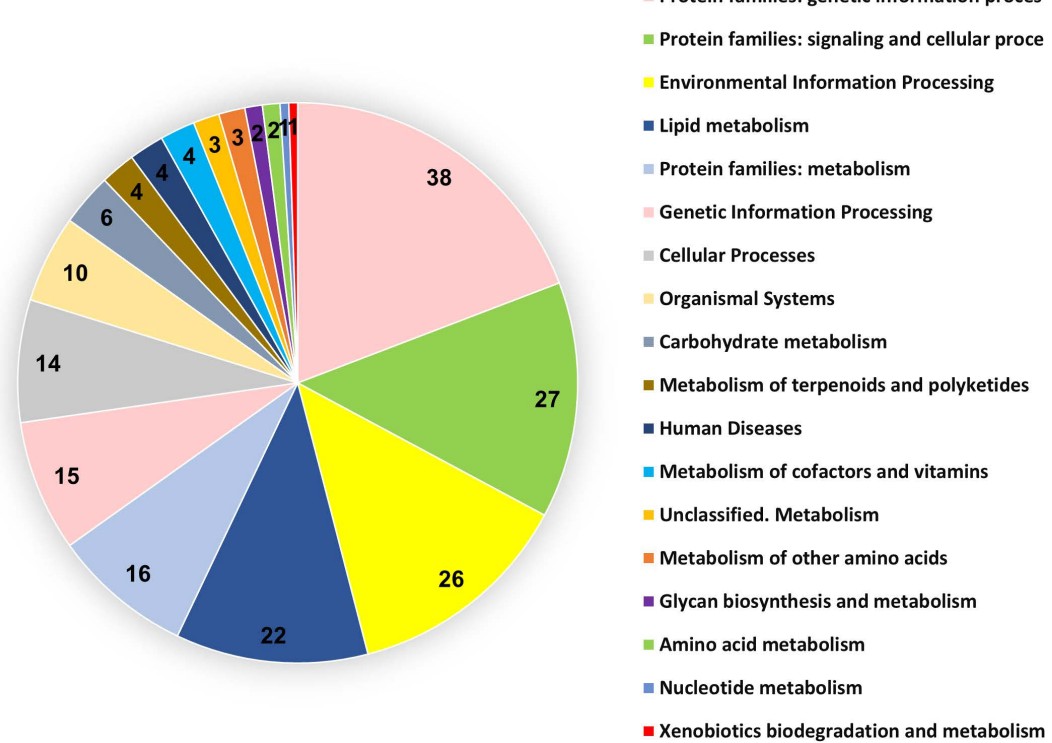

## BlastKOALA: Functional categories

**Legend:**
- Protein families: genetic information proces
- Protein families: signaling and cellular proce
- Environmental Information Processing
- Lipid metabolism
- Protein families: metabolism
- Genetic Information Processing
- Cellular Processes
- Organismal Systems
- Carbohydrate metabolism
- Metabolism of terpenoids and polyketides
- Human Diseases
- Metabolism of cofactors and vitamins
- Unclassified. Metabolism
- Metabolism of other amino acids
- Glycan biosynthesis and metabolism
- Amino acid metabolism
- Nucleotide metabolism
- Xenobiotics biodegradation and metabolism

**Fig 4. Gene ontology analyses of 490 identified gland genes.** BlastKOALA KEGG pathway analysis was performed on taxonomy group Eukaryotes, Animals, Arthropods (Taxonomy ID 7070) searching KEGG databases family_eukaryotes.pep + genus_prokaryotes.pep. 198 genes (40%) could be analyzed (S6 Table) and were put into 18 functional categories.

phase of the iBeetle screen (iB_01701), but no stink gland phenotype was observed (https://ibeetle-base.uni-goettingen.de/details/TC010489), since in the larval injection screen the larvae died or adult eclosion did not happen, while in the pupal injection screen the inspection for stink gland phenotypes was not carried out [32]. The larval or pupal death after RNAi-mediated knock-down of *Tc_lac2* had also been observed, when its function for the oxidation of precursors to generate *ortho*-benzoquinones used in cuticle sclerotization and pigmentation was identified [41,42].

Laccase2 represents a typical multicopper oxidase [40] and can oxidize *ortho*- and *para*-diphenols to their corresponding benzoquinones [48,49]. For cuticle sclerotization and tanning Laccase2 catalyzes the formation of *ortho*-quinones by oxidizing catechols such as, 3,4-dihydroxyphenylalanine (DOPA), dopamine, N-beta-alanyldopamine, and N-acetyldopamine [42]. In the odoriferous stink glands, harmless *para*-diphenols enter the head of the cuticle-lined so-called 'vesicular organelles', which are used as reaction chambers within the gland cells, where the final oxidation step to toxic *para*-benzoquinones happens [17]. Here, we show that this phenoloxidation requires Laccase2, since the knock-down of *Tc_lac2* causes a *para*-benzoquinone-less and colorless stink gland phenotype (Fig 5B-5D). Also, when only the major splice variant *Tc_Lac2A* is targeted for knock-down, a *para*-benzoquinone-less and colorless stink gland phenotype has been observed. However, when only the minor splice variant (*Tc_Lac2B*) was targeted, the stink glands appeared partially melanized and the volatile *para*-benzoquinones were only reduced, which also explains, why in the iBeetle screen for Tc_010490 (iB_12548) not "colorless" but "darker" and no quinone-less phenotypes were recognized. A difference in

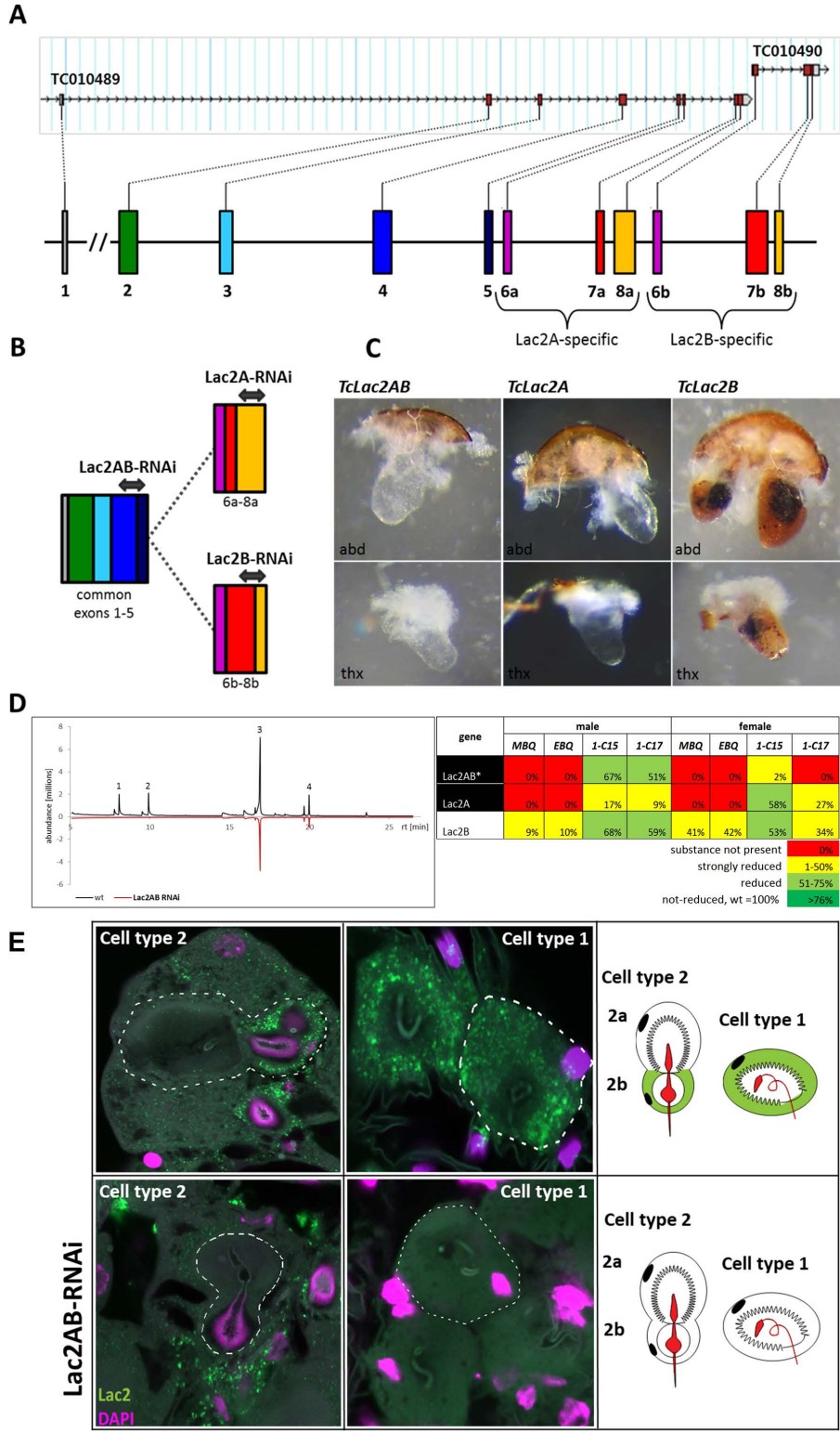

**Fig 5. Characterization of the stink gland function of the phenoloxidase Laccase2. (A)** Gene structure of *Tc_lac2*. In iBeetle-Base, *Tc_lac2* is still covered by two independent gene models: Tc_010489 and Tc_010490. However, Tc_010490 actually covers only some alternatively spliced Exons of *Tc_lac2*, while Tc_010489 covers the complete gene and the first set of alternatively spliced Exons. At NCBI, *Tc_lac2* is presented as one gene with

alternative Exons, which we display here as colored bars and mark the alternative exons with a and b (as in [41]). **(B)** For knock-down of common exons or differently spliced ones, selected *lac2* RNAi fragments were chosen as indicated by the double-headed arrows: To target both splice variants dsRNA was produced from a sequence spanning the border of common exons 4 and 5 (*Lac2AB*); to target the different splice variants, RNAi fragments *Lac2A* and *Lac2B* were constructed in regions with highest sequence difference (exon 8a for *Lac2A* and spanning exon border 7/8 for *Lac2B*, respectively. **(C)** Morphological stink gland phenotypes induced by *Tc_lac2* knock-down: when knocking down all transcripts (*Tc_Lac2AB*) or the transcripts of TC_010489 (*Tc_Lac2A*) both abdominal and thoracic glands showed a colorless phenotype, whereas knock-down of the Tc_010490 transcripts (*Tc_Lac2B*) caused a partially melanized gland content, similar to the "darker" phenotype recognized in the 3rd phase of iBeetle (S1 Table). **(D)** Also the stink gland content composed of the four main volatile substances 2-Methyl-1,4-benzoquinone (peak 1; MBQ), 2-Ethyl-1,4-benzoquinone (peak 2; EBQ), 1-Pentadecene, (peak 3; 1-C15), and 1-Heptadecene (peak 4; 1-C17) was differentially affected by the knock-downs: While knocking down all transcripts (*Tc_Lac2AB*) or the transcripts of Tc_010489 (*Tc_Lac2A*) caused a *para*-benzoquinone-less phenotype, knock-down of the Tc_010490 transcripts (*Tc_Lac2B*) resulted only in a reduction of the *para*-benzoquinones in abdominal stink glands. Besides the complied measurement results, also the GC-MS chromatogram for abdominal glands of wild-type male beetles (black line positive values) versus knock-down beetles using *Lac2AB* RNAi is shown (red line, for better comparison plotted as negative values), which also indicates the complete lack of *para*-benzoquinones (peaks 1 and 2) while the alkenes (peaks 3 and 4) seem only reduced. **(E)** Fluorescence *in situ* hybridization (FISH) performed on abdominal glands of ten-day old wild type or *Lac2AB*-knock-down beetles. Expression patterns detected by probes against *Lac2AB* (green) are also depicted in schemes of the two different cell types 1 and 2 (expression restricted to cell 2b; [17]). Specificity of the probe is verified by the lack of cytoplasmic fluorescence in the gland tissue of *Lac2AB* RNAi-treated beetles. DAPI staining (magenta) was used to identify the cell nuclei but also stains part of the vesicular organelle in cell type 2b. In the schemes, the vesicular organelle is outlined in red. Cell outlines are indicted by dashed lines. Green fluorescence outside the cells is due to unspecific trapping of the probe or reagents.

the phenotypes caused by the splice-variant-specific gene knock-down was also observed for cuticle tanning, with the *Tc_Lac2A* knock-down causing the identical phenotype to the knock-down of both transcript variants (*Tc_Lac2AB*), while *Tc_Lac2B* contributes little to cuticle tanning but was nonetheless indispensable for survival [41]. Although the knockdown effect of the more abundant variant *Tc_Lac2A* was clearly stronger, as stink gland secretions were colorless and completely lacked the *para*-benzoquinones compounds, both isoforms are needed for proper biosynthesis of secretion, with *Tc_Lac2A* being necessary for the oxidation of *para*-benzoquinone precursors and an unclear role for *Tc_Lac2B*. Recently, *laccase2* was functionally analyzed also in another coleopteran species, the cotton boll weevil *Anthonomus grandis* [50], with RNAi-mediated knock-down causing similar cuticle sclerotization and tanning phenotypes, however, without being able to identify adult secretory phenotypes, since all knock-down weevils died with severe soft cuticle phenotypes. The pleiotropic use of *Tc_lac2* in *Tribolium* for both formation of *ortho*-quinones involved in cuticle tanning as well as for *para*-benzoquinone production in the defensive stink glands is in contrast to findings in rove beetles [13] and termites [51], where paralogous clade-specific laccases were identified to catalyze the production of *para*-benzoquinones in defensive tergal glands [13] or through autothysis allowing the crystalized enzyme to activate separately stored precursors [51], respectively.

Expression profiles based on RT-PCR analysis of RNA prepared from prepupal, pupal, and adult stages of the *T. castaneum* indicated that *Tc_Lac2A* transcripts are more abundant than *Tc_Lac2B* transcripts at almost all stages [41], which is consistent with our transcriptomics analysis of the stink glands, that indicates that *Tc_Lac2A* transcripts are at least 100x more abundant than *Tc_Lac2B* transcripts in the stink gland tissues (S2 Table). To detect, where in the gland tissue *Tc_lac2* is expressed, we performed whole mount fluorescence *in situ* hybridization (FISH) of abdominal stink glands. *Tc_lac2* transcripts are detected in both secretary cell types 1 and 2, with the expression being specific to cell 2b of cell type 2 (Fig 5E). Type 1 cells are individual or paired cells that are present over the whole surface of the reservoir with each cell harboring a huge vesicle, which itself carries a simple cuticle-lined organelle, while type 2 cells aggregate into lobules and are composed of two fused cells 2a and 2b, which also carry in their vesicles a cuticle-lined organelle that is divided into a head lying inside the vesicle of cell 2a and a bulb almost filling the vesicle of cell 2b ( [17]; Fig 5E). To produce Laccase2 only in cell 2b of type 2 cells, which represents the last secretory cell in contact with the vesicular organelle before releasing the content via tubules into the reservoir, is consistent with the required self-protection of the gland cells from the toxic *para*-benzoquinones, which are made by oxidizing harmless *para*-diphenols in the cuticle-lined organelles [17].

## Conclusion

*T. castaneum* presents a significant worldwide pest of stored grains that not only feeds on it but also produces toxic secretions, which render the grains useless and hazardous to human health [30]. By a genome-wide RNA interference screen (iBeetle; [32,35]) and the re-analysis of gland-specific transcriptomics [38] based on a significantly improved genome annotation [34], we identified a large set of 490 genes being involved in odoriferous stink gland function. Only about 40% of these genes have molecularly characterized homologs in the vinegar fly, 213 genes seem to have no fly homolog and for 13 genes no gene ontology at all could be identified. Thus, the genome-wide gene identification for the function of a specific tissue that is not present in any of the key model organisms appears to be very rewarding, since many of the identified genes might be involved in still uncharacterized metabolic pathways or in self-protection against the produced and stored defensive, toxic secretions. The characterization of these genes and their function will be exciting even though not as straight forward as for known homologous genes.

Moreover, we could identify Laccase2, which had previously been described to be involved in the generation of *ortho*-benzoquinones used in cuticle sclerotization and pigmentation [41,42], as the phenoloxidase involved in generating also the toxic *para*-benzoquinones in the stink glands of the beetle. Laccase2 is produced in the last secretory cell in contact with the vesicular organelle before releasing the content via tubules into the gland reservoir. This way, the gland cells are self-protected from the toxic *para*-benzoquinones, which are finally made only in the cuticle-lined organelles [17]. The use of laccase homologs as phenoloxidases for *para*-benzoquinone synthesis in diverse insect clades indicates that there is a constrained chemical space for the production of defensive secretions resulting in convergent evolution of different defensive glands or mechanisms such as autothysis [13]. However, the evolution of the biosynthetic pathways resulting in defensive glands producing *para*-benzoquinones appears to have followed different routes in the different clades. In the rove beetles, gene duplications for involved enzymes and recruitment to a novel gland cell type are observed [14], which enables paralogous genes to escape from a potential pleiotropic conflict arising from optimization selection for the new biosynthesis purpose [1]. This is in contrast to the here demonstrated pleiotropic use of *Tc_lac2* for both cuticle tanning as well as the production of defensive secretion, which follows the model of partial enzyme pathway co-option into a novel gland cell type [1]. This indicates that evolution can use diverse routes to converge on analogous structures or biosynthetic pathways.

## Materials and methods

### *Tribolium* rearing

*T. castaneum* (Herbst, 1797; Insecta, Coleoptera, Tenebrionidae) was reared on organic wheat flour supplemented with 5% yeast powder at 28°C and 40% relative humidity under constant light [15]. In the 3rd phase of the iBeetle screen, dsRNA injection was performed on transgenic line pBA19 (also called Pig-19; [52]).

### Screening procedure of iBeetle 3rd phase

In the 3rd phase of the iBeetle screen, 4,520 genes were covered. dsRNAs corresponding to the original iBeetle-fragments (iBeetle-Base; [37] were commercially (Eupheria Biotech GmbH, Dresden, Germany) obtained [32]. For resuspension, injection buffer (10x stock: 40 mM KCl, 0.3 mM $KH_2PO_4$, 0.7 mM $Na_2HPO_4 \cdot 2H_2O$, 14 mM NaCl) was used. dsRNA was injected with a concentration of 1 µg/µl into mid-pupal stages of male and female beetles [33,35]. Adult stink glands were dissected and morphologically analyzed under a stereomicroscope at day 21 after eclosion. Altered stink gland phenotypes were documented as previously described [33] by photography (Leica microscope MZ16FA, Q-imaging camera, GmbH, Wetzlar, Germany) and recorded in iBeetle-Base [37].

## Gas chromatography-mass spectrometry analysis

Glands from three ten-day old RNAi-treated adults were analyzed by semi-quantitative gas chromatography-mass spectrometry (GC-MS) independently for the thoracic and abdominal glands to detect the known main gland volatiles as previously described [33].

## Transcriptomic analysis

To re-analyze the previously published gland transcriptomic data [38], we used the latest version of the *T. casteneum* transcriptome "OGS3_mRNA.fasta" and selected for each transcript the longest isoform, which provided us with 16,593 unique transcripts [34]. From the stink-gland-specific transcriptomes generated in Li et al. [38], we included the following samples: female abdominal glands, female prothoracic glands, male abdominal glands, male prothoracic glands, and the control sample coming from anterior abdomen of wild type beetles. We mapped each sample to the current transcriptome, using Bowtie2 [53] with default parameters in -local mode. To make the analysis as comparable as possible, we calculated the reads depth as described in Li et al. [38], namely as: (# of reads mapped to the transcript * 38)/ length of transcript (in base pairs). In cases, where 0 reads mapped to a transcript, we added 1 previously to this step, to not lose the information about the transcripts in the next step of calculating logFC. We calculated the log2FC between each gland sample and the control sample (anterior abdomen) using the formula: log2 (reads mapped to transcript in sample/ reads mapped to transcript in control) according to Li et al. [38]. We defined all transcripts that showed FC >= 16 (log2FC >= 4) as 'gland specific' (S2 Table, Sheet 1). To compare our analysis to the datasets of Li et al. [38], we filtered their dataset according to our parameters, including adding +1 to cases where no reads mapped to the sample (S2 Table, Sheet 2). All described calculation steps, figures and calculation of overlap were analyzed with R, version 3.6.1 ("R: The R Project for Statistical Computing").

## Homolog search and gene ontology analysis

The closest *D. melanogaster* homologs were documented (S6 Table) as indicated by iBeetle-Base [37]. To analyze gene ontologies of the identified 490 genes and potentially recognize metabolic pathways, we performed ShinyGO [44], BlastKOALA [45], and eggNOG-mapper [46] analyses. For ShinyGo the respective 490 Tc numbers (S1 File) were entered online (http://bioinformatics.sdstate.edu/go/) to search the KEGG pathway data base for the species *T. castaneum* (FDR cut off 0.05). For BlastKOALA and eggNOG-mapper the 490 encoded protein sequences were provided in FASTA format (S2 File) and entered online at https://www.kegg.jp/blastkoala/ or http://eggnog-mapper.embl.de/, respectively. The respective results are provided in (S6 Table). To generate the column chart for S3B Fig, the KEGG orthologies derived from the eggNOG analysis were summarized online (https://www.genome.jp/kegg/ko.html) using the KEGG ORTHOLOGY (KO) Database [45].

## dsRNA generation for *laccase2* knock-downs

To specifically knock-down the complete gene *Tc_lac2* gene or the different splice variants the following primers were used: *Lac2AB* (Tc_Lac2AB_F1: ATGCACGAAGACGCTACTGA and Tc_Lac2AB_R1: TATGCCACACTCCCCTAAGC); *Lac2A* (Tc_Lac2A_F1: AACTCGAATCCTAACCTCGTT and Tc_Lac2A_R1: ATGCAAGCGCCATATTGTA); and *Lac2B* (Tc_Lac2B_F1: AGATCGCCTGACCAGAATGT and Tc_Lac2B_R1: TGATATAGGTGGCAGATGGTTC). The PCR-amplified fragments were cloned into pJET1.2/blunt vector (Thermo Fisher Scientific) and after sequence verification re-amplified with T7-overhang primers (pJET1.2_F_up_T7: ACACTTGTGCCTGAACACCATATC and pJET1.2_ +T7_R: TAATAC-GACTCACTATAGGAAGAACATCGATTTTCCATGGCAG). To generate dsRNA, the purified PCR product was used as template for *in vitro* transcription using the MEGAscript T7 Kit from Ambion (Life Technologies GmbH, Darmstadt, Germany).

## Fluorescence *in situ* hybridization of stink gland tissue

RNA probes for *Tc_lac2* were prepared from purified PCR products. *In vitro* transcription was performed using the Fluorescein RNA Labelling Mix (Roche Applied Science, Mannheim, Germany). A mixture of 0.1% Tween20, 50% formamide, 20 µg/ml heparin, and 5x SSC (20x stock: sodium citrate 300mM, 3M NaCl, pH 5.5 adjusted by citric acid) pH 4.5 was used for dissolving the RNA probes. For fluorescence *in situ* hybridization on whole mount stink glands, around twenty abdominal and prothoracic stink glands from adult beetles were isolated along with a small part of the exoskeleton and attached to a 12 well-plate with sylgard bottom (World Precision Instruments, Berlin, Germany) with a pin. Subsequently, phosphate-buffered saline (1x PBS) was added to gland containing wells. For fixation, the glands were put in 4% of paraformaldehyde (PFA, Sigma-Aldrich Chemie GmbH, Munich, Germany) in PBS at 8°C for 2-2.5 h or at 4°C overnight. Glands were rinsed and washed two times in 50% methanol/PBX (0.03% TritonX-100 in 1x PBS). For antigen retrieval, the glands were incubated with proteinase K solution (0,5 µl in 500 µl PBT) for 5 min. After rinsing and washing glands in PBT for 10 min, the gland tissues were post-fixed at room temperature in 4% PFA for 30 min. Glands were again rinsed and washed for 5 min four times with PBT. Afterward, 50% pre-warmed Hyb-buffer (0.03% TritonX-100, 50% Formamide, 5x SSC pH 5.5, heparin 100 µg/ml, 100 µg/ml Yeast RNA, 100 µg/ml salmon sperm DNA) in PBT was used to washed the glands for 5 min at 60°C. In the next step, the gland tissues were placed for incubation in Hyb-buffer at 60°C for 3 hours. RNA probes were diluted to 100 ng/ml in Hyb-buffer and kept at 95°C for 2 min and then instantly put for 10 min on ice. Subsequently, glands were placed in 1 ml probe solution (per well) and put at 60°C overnight. Next day, 2x pre-warmed SSC was used to rinse and for two washes at 60°C for 10 min. Washing buffer (2x SSC, 0.03% TritonX-100, 50% Formamide) was added to the gland tissues and put for 45 min at 60°C. Afterwards, washing buffer-TBST 50% (0.03% TritonX-100 in TBS) was added and placed at 60°C for 10 min, then in TBST for 10 min at 65°C, and at RT in TBST for 10 min. Blocking solution (1:10 in TBST, Roche Applied Science, Mannheim, Germany) was added to the gland tissues at 8°C for 3 h. In the next step, the gland tissues were incubated overnight at 8°C in streptavidin (1:200 in blocking solution) and Hoechst 33342 (1 mg/ml stock 1:1000, Sigma-Aldrich Munich, Germany). On the morning of the third day, after 4 times washing in TBST, staining solution was added to the glands. (200 amplification diluent + 4 µl tyramides; TSA fluorescein detection kit, Cat#: NEL701A001KT, PerkinElmer, USA). Gland tissues were incubated for 90 min at RT in the dark. After incubation, the tissues were rinsed in TBST and four times washed for 10 min in TBST. Glands were put on a microscopic glass slide and Mowiol 4–88 (Sigma-Aldrich GmbH, Munich, Germany) was applied over them. The slides were kept at 8°C overnight before microscopic analysis. Photographs were taken with a Zeiss LSM780 confocal laser microscope. ZEN 3.2 software was used to adjust brightness/contrast of captured pictures. Adobe Photoshop (CS5) and Adobe Illustrator (CS5) were used to draw sketches of abdominal and thoracic gland cells.

## Supporting information

**S1 Fig. Comparision of the different transcriptomics analyses.** Orange ellipse: 208 stink gland genes identified by Li et al. [38] (OGS2: Groups 1, 2, 3, 5, 9, 10). Blue ellipse: 75 stink gland genes identified by transcriptome re-analysis with OGS3 [34]. Red ellipse: 60 stink gland genes identified by transcriptome re-analysis as performed for OGS3, but with the gene set (OGS2) available at the time of Li et al. [38]. Altogether, the three transcriptomics analyses identified 249 genes with stink gland-specifically enhanced expression (S3 Table).
(PDF)

**S2 Fig. KEGG pathways [54] identified by ShinyGO and BlastKOALA analyses.** (A) ShinyGO analyzed 438 genes (89%) (S6 Table) and found "SNARE interactions in vesicular transport" as the only significantly enriched KEGG pathway. The five genes Tc_001666, Tc_008320, Tc_009870, Tc_015165, Tc_015183 encoding blocked early in transport 1 (BET1), vesicle-associated membrane protein 7 (VAMP7), syntaxin 16 (STX16), golgi SNAP receptor complex member 1 (GOS1), and vesicle-associated membrane protein 2 (VAMP2), respectively, are all involved in SNARE function (ko04130). In the

BlastKOALA analysis, "SNARE interactions in vesicular transport" (ko04130) was identified with the same five genes. In addition (B-D), the following three KEGG pathways were identified: "Lysosome" (ko04142) (B) with 17 genes: Tc_005432 (cathepsin B; CTSB); Tc_008203 (Niemann-Pick C2 protein; NPC2); Tc_008780 (glucosylceramidase; GBA; Tc_008912 (AP-3 complex subunit sigma AP3S); Tc_012600 (octopamine receptor beta; Octbeta); Tc_015151 (arylsulfatase B; ARSB); Tc_015811 (lysosomal acid phosphatase; ACP2); Tc_016314 (formylglycine-generating enzyme FGE; SUMF1); Tc_033512 (G protein-coupled receptor); Tc_034418 and Tc_034419 (TSPAN30; CD63 antigen); as well as Tc_007186, Tc_012838, Tc_012841, and Tc_034670 encoding lysosomal acid lipases/cholesteryl ester hydrolases (LIPA); as well as Tc_014388 and Tc_034847 encoding Major Facilitator Superfamily (MFS) transporters (SLC17A); "Pancreatic secretion" (ko04972) (C) with eight genes: Tc_005635, Tc_007019, Tc_015344, and Tc_030065 (trypsin; PRSS1_2_3); Tc_011288 (potassium large conductance calcium-activated channel subfamily M alpha member 1; KCNMA1); Tc_014033 (sodium/potassium-transporting ATPase subunit beta; ATP1B); Tc_032365 (classical protein kinase C alpha type; PRKCA); Tc_033681 (ovochymase; OVCH); and "MAPK signaling pathway – fly" (ko04013) (D) with six genes: Tc_000385 (mitogen-activated protein kinase kinase 7; MAP2K7); Tc_004593 (dual oxidase; DUOX); Tc_034116 (Runt); Tc_012298 (E3 ubiquitin-protein ligase SIAH1); Tc_014568 (TRAF2 and NCK interacting kinase; TNIK); Tc_031200 (fyn-related kinase PTK5; FRK).
(PDF)

**S3 Fig. BRITE analyses of KEGG orthologies.** The number of genes that have been assigned to the different KEGG orthology pathways in the Brite analyses of BlastKOALA (A) and eggNOG (B) are presented as column charts. While the BlastKOALA KEGG pathway analysis covered only 198 genes (40%), eggNOG-mapper analyzed 365 genes (75,5%) for the 490 provided query proteins (S6 Table). Similar to the BlastKOALA Brite analysis (A), also the eggNOG-mapper Brite analysis (B) identified many enzymes including kinases, phosphatases, glycosyltransferases, peptidases, and cytochrome P450s as well as transporters and other proteins involved in membrane trafficking. However, also a large number of transcription factors were recognized.
(PDF)

**S1 Table. Morphology and GC-MS analysis of stink gland volatile compounds in knock-down beetles of 178 identified genes in 3rd phase of iBeetle.** The genes are listed by the iBeetle-number with genes also identified in transcriptomics analyses indicated in bold (color code respective to Fig 3). The morphological phenotype depiction is given in Fig 1 (color code respective to Fig 1I). In the GC-MS analysis, the analyzed substances were 2-Methyl-1,4-benzoquinone (MBQ) and 2-Ethyl-1,4-benzoquinone (EBQ), as well as 1-Pentadecene (1-C15) and 1-Heptadecene (1-C17). Column L (Venn Fig 2A) indicates the color code for extraction of the data to generate Fig 2A. The further analyzed gene *laccase2* (iB_12548; Tc_010490) is indicated by italics and grey background.
(XLSX)

**S2 Table. Stink gland transcriptomics re-analysis based on gene sets OGS3 (A) and OGS2 (B).**
(XLSX)

**S3 Table. Expression data of 249 transcriptomics-identified genes highly expressed in stink gland tissue.** The genes are listed by the Tc-number grouped according to the color code of S1 Fig, with the six genes also identified in iBeetle indicated in bold (color code respective to Fig 3).
(XLSX)

**S4 Table. Expression data of 178 iBeetle 3rd phase-detected genes involved in stink gland function.** The genes are listed by the iBeetle-number with the five genes also identified in transcriptomics analyses indicated in bold (color code respective to Fig 3). The further analyzed gene *laccase2* (iB_12548; Tc_010490) is indicated by italics and grey background.
(XLSX)

**S5 Table. Morphology and GC-MS analysis of stink gland volatile compounds in 41 genes identified by transcriptome re-analysis.** The genes are listed by the Tc-number with gene Tc_033013 also identified in iBeetle indicated in bold. The morphological phenotype depiction is given in Fig 1 (color code respective to Fig 1I). In the GC-MS analysis, the analyzed substances were 2-Methyl-1,4-benzoquinone (MBQ) and 2-Ethyl-1,4-benzoquinone (EBQ), as well as 1-Pentadecene (1-C15) and 1-Heptadecene (1-C17). Column L indicates the color code for extraction of the data to generate Fig 2B.
(XLSX)

**S6 Table. Gene ontology of 490 iBeetle- and transcriptomics-identified genes involved in stink gland function.** The genes are listed by the Tc-number. The 249 genes recognized by the transcriptomics approaches are color coded respective to S1 Fig, with the six genes also identified in iBeetle indicated in bold (iBeetle number color code respective to Fig 3). The closest *D. melanogaster* homologs are denoted as indicated by iBeetle-Base [37]. In addition, gene ontology analyses were carried out using ShinyGO [44], BlastKOALA [45], and eggNOG-mapper [46]. In ShinyGO, n.a. specifies that genes were not analyzed, whereas SNARE indicates the five genes involved in "SNARE interactions in vesicular transport". In BlastKOALA, 292 genes were not analyzed, while in eggNOG-mapper only for 125 genes no orthologs were identified. For genes screened in iBeetle, the phase of the screen is provided in the far-right column. The further analyzed gene *laccase2* (Tc_010490; iB_12548) is indicated by italics and light grey background.
(XLSX)

**S1 File. List of 490 TC numbers analyzed in ShinyGO.**
(XLSX)

**S2 File. FASTA file of 490 proteins analyzed in BlastKOALA and eggNOG-mapper.**
(TXT)

## Acknowledgments

We would like to thank the Dept. of Forest Zoology and Forest Conservation as well as the Dept. of Crop Sciences Göttingen, Georg-August-University Göttingen, for allowing us to use their GC-MS analysis equipment and for advice.

## Author contributions

**Conceptualization:** Gregor Bucher, Ernst A. Wimmer.

**Data curation:** Elisa Buchberger.

**Formal analysis:** Elisa Buchberger.

**Funding acquisition:** Gregor Bucher.

**Investigation:** Bibi Atika, Sabrina Lehmann, Musa Dan'azumi Isah, Zahra Basirkazerouni.

**Methodology:** Michael Rostás.

**Project administration:** Ernst A. Wimmer.

**Supervision:** Michael Rostás, Ernst A. Wimmer.

**Validation:** Elisa Buchberger.

**Writing – original draft:** Bibi Atika, Ernst A. Wimmer.

**Writing – review & editing:** Sabrina Lehmann, Elisa Buchberger, Gregor Bucher, Ernst A. Wimmer.

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
