## [Decision Letter · Decision Letter 0]

21 Apr 2025

PGENETICS-D-25-00083

Genome-wide identification of genes involved in beetle odoriferous defensive stink gland function recognizes Laccase2 as the phenoloxidase responsible for toxic para-benzoquinone synthesis

PLOS Genetics

Dear Dr. Wimmer,

Thank you for submitting your manuscript to PLOS Genetics. After careful consideration, we feel that it has merit but does not fully meet PLOS Genetics's publication criteria as it currently stands. Therefore, we invite you to submit a revised version of the manuscript that addresses the points raised during the review process.

In particular, as the reviewers point out, reanalyses of the RNAseq data according to the latest TriCast1.1 genome assembly should be completed. Further, integration of more relevant literature and more careful proof-reading would be desirable. 

Please submit your revised manuscript within 60 days. If you will need more time than this to complete your revisions, please reply to this message or contact the journal office at plosgenetics@plos.org. Please include the following items when submitting your revised manuscript:

We look forward to receiving your revised manuscript.

Kind regards,

Kenneth V. Halberg, PhD

Guest Editor

PLOS Genetics

Giovanni Bosco

Section Editor

PLOS Genetics

Aimée Dudley

Editor-in-Chief

PLOS Genetics

Anne Goriely

Editor-in-Chief

PLOS Genetics

**Journal Requirements:**

- ® on pages: 19, 20, and 21.

4) We notice that your supplementary Figures are included in the manuscript file. Please remove them and upload them with the file type 'Supporting Information'. Please ensure that each Supporting Information file has a legend listed in the manuscript after the references list.

Potential Copyright Issues:

i) Please confirm (a) that you are the photographer of Figures: 1 and 5, or (b) provide written permission from the photographer to publish the photo(s) under our CC BY 4.0 license.

**Reviewers' comments:**

Reviewer's Responses to Questions

**Comments to the Authors:**

Reviewer #1: This is a detailed study which reveals interesting and important insights into the function of insect defensive glands. Results of a large systematic study are presented, for which the data appears to be rigorous. I also think it is an important message that 60% of the stink gland genes are not found in Drosophila. For historical reasons Drosophila often acts as a kind of reference insect, and this is another important reason for research such as this, which is extending other insects e.g. Tribolium as model species.

I feel like the clarity and quality of the paper could be improved through some careful rewriting. I found there is quite a lot of detail, particularly in the results part, which could be streamlined, along with some re-writing to make sure the message is always clear and the significance and relation to other studies comes through more strongly. The quality of the writing and use of English could do with improving in quite a few places. For example line 77 ‘secretion’s’, line 79 capitalisation after semicolon, line 187 ‘in view cases’, comma in line 115, line 143 simplify this sentence, line 157-158 the use of English is quite strange. In general I think careful proofreading could improve the paper. There seem to be a couple of inconsistencies such as using Tribolium or T. castaneum, and in gene names e.g. Tc0… or Tc_0…

I find this point about different classes of genes interesting, and maybe you could discuss this a bit further. i.e. some are specific to the stink gland, whilst some are expressed more widely but still have an important functional role. I see for example that the gene TC030914/ATP7 also has some enriched expression in the free Malpighian tubule according to BeetleAtlas, so maybe there is a common role in secretory cells?

I feel like on line 102 you could do with mentioning e.g. the Brückner, 2021 paper, which provides important insights into a beetle defence gland, and you can highlight how Tribolium can extend or add a useful comparison with what has been found in other insect species. I also feel like some more comparative discussion about findings in other species would be nice at the end of your paper to place your findings in context, and maybe draw out what similarities or differences are emerging.

A couple of small points:

in some places e.g. line 108 you refer to organelle – is this the right term? You seem to be referring to a structure external to the cell? To help with this point, and in general to make your findings clearer it could be nice to have a summary figure labelling the relevant structures and adding the key molecules or pathways.

Figure 5E – add labels to easily interpret what is being shown in green and blue.

some places e.g. line 126 it is not clear what you mean by ‘gene models’.

line 114 – make it clear that sclerotization refers to toughening of the cuticle, as PLOS Genetics has a broad readership.

line 287 – can you comment on whether any further genes are found in any of the other new Tribolium castaneum genome versions that have been released?

I assume your raw RNA-Seq data is already deposited to a database, but I wonder whether it would also be useful to deposit you new counts data made against the updated genome?

Reviewer #2: This is a very interesting manuscript describing an RNAi screen and transcriptomics analyses to identify candidate genes that function in the stink gland of the flour beetle Tribolium. The work made use of improved genome assembly and annotations to follow up on their initial screens and identified a reasonable group of genes that may function in development of the glands and in the reactions for producing the protective quinones. The work is thorough and carried out with appropriate experimental designs and statistical analyses.

It is quite interesting that for a large number of the genes of interest, there is no homolog in Drosophila, illustrating a case in which Drosophila is not useful as a model species. The RNAi results combined with chemical analyses demonstrate convincingly that laccase-2 is the oxidase responsible for oxidation of benzophenols to corresponding quinones, and that laccase-2 is localized in the expected tissue and developmental timing. The analysis of the expression and function of splicing isoforms of laccase-2 are also quite interesting. Laccase-2 is not expressed with great specificity in the stink glands because it also functions in cuticle tanning and sclerotization, where it is secreted from epidermal cells after the molt, so serves different oxidative functions.

This is a very well written paper and should be of interest for a wide group, including chemical ecology, biochemistry, and insect molecular biology.

I have suggestions for only a few minor revisions:

It seems to me that Fig 2D is an important part of the results but is presented in a small format that does not highlight its significance. I think it might be presented separately as a more prominent figure.

line 317: spelling of lysosome

It would be appropriate to cite Arakane et al. "Laccase 2 is the phenoloxidase gene required for beetle cuticle tanning." Proc Natl Acad Sci U S A. 2005 Aug 9;102(32):11337-42. doi: 10.1073/pnas.0504982102 as a key paper on discovery of laccase-2 function in cuticle of Tribolium.

Reviewer #3: Atika et al. presents a comprehensive profiling of gene candidates involved in the production of toxic para-benzoquinones in Tribolium castaneum defensive glands. By reanalyzing a previous transcriptome (Li et al., 2013) and a more extensive genome-wide RNAi screen, they identified a one-third increase in mapping rate, and a two-fold increase in identified genes differentially expressed in the defense gland. Compared to the original study, this work aimed to use improved gene models in the most recent genome annotation, more detailed RNAi phenotypic screen (gland dissection) and relaxed fold-change cutoff, while maintaining the same morphological categories (as in Li et al., 2013) to characterize gland phenotypes as a result of gene-specific RNAi effects. The authors identified concordant and newly annotated genes with stink gland-restricted expression, with a specific highlight of a phenoloxidase Laccase2 in para-benzoquinone synthesis, which was missed in the original transcriptome study.

Overall, I find that the study placed much effort in detailing the gene mining process. The strength of this study lies in the much-needed, updated view of the genes differentially expressed in the stink gland of T. castaneum, which is expected of the past decade of improved convenience of sequencing technologies, alongside significant efforts in phenotypic screens by conducting gland dissections. However, I found that the study falls short of providing some improvements in the transcriptome analysis, and point out several points for improvement in the analysis of transcriptomic data. I also suggest the authors refer to prior work in the literature on laccases in beetles, including in benzoquinone synthesis.

Major comments

1) Overall, this paper does not frame the work sufficiently in a broader biological context, making it quite arduous to read. What is interesting/striking/unexpected about the findings in this paper are not articulated well. It reads instead like a resource paper, devoid of much biological significance. I was also struck that the paper did not mention the detailed recent work on rove beetles that demonstrated a pathway for benzoquinone synthesis that includes a laccase (which again carries out the terminal step). In this case it is not laccase2 but a novel duplicate. See two recent papers: 10.1016/j.cell.2021.11.014, 10.1016/j.cell.2024.05.012

The authors could also discuss the new findings in the context of the previously characterized roles of laccase2 in T. castaneum (10.1073/pnas.0504982102), which appears to be conserved in insects. There are no evolutionary inferences in this paper to do with the co-option/pleiotropy of laccase2. The authors need to frame the findings of the paper in the context of these studies.

2) The phenotypic characterization does not seem intuitive and requires revision. Comparisons of different categories are not very clear. I suggest adding scale bars, especially to clarify size differences that are not obvious in III and VII. Also, comparison of colors would maybe require similar lighting contrast in all panels, which is not the case at least in VI (too light) and VII (too dark).

In Figure 1,

- III appears slightly damaged and irregularity in size is not immediately obvious. If possible I suggest replacing this example.

3) Reanalysis of transcriptome data should include the log adjusted p-values for including significance values. From their reanalysis of transcriptome data, they reduced the log2 fold cutoff from 8 to 4 and highlighted that it allowed them to identify lac2. While it is expected that reducing the cutoff can result in fewer false negatives, especially given the not-so-well mapped sequencing data, presenting the data in terms of both log2 fold change and p-value significance, in the form of a volcano plot will be a better way forward.

3) Gene ontology analyses are not very informative. I don’t find Figure 4 particularly informative, especially with “Human Diseases” as a functional category in this context. I suggest that the authors proceed with more caution and reduce descriptive emphasis using gene ontology and kegg pathway analyses, instead of using them as factual descriptions of underlying biology in transcriptomic data.

4) Isoform-specific roles of laccase2 in benzoquinone synthesis

The authors reported that “when only the minor splice variant (Tc_Lac2B) was targeted, the stink glands appeared partially melanized and the volatile para-benzoquinones were only reduced” (line 385), then went on to mention that Tc_Lac2B “contributes little to cuticle tanning but was nonetheless indispensable for survival”. I found this to have digressed from the original point on the roles of laccases in gland-specific functions, and instead recommend the authors to address the melanized phenotype of gland secretion.

In addition, I urge the authors to validate using RT-PCR the RNAi knockdown of isoform-specific laccase2, especially with previously reported work that showed unintended reduction of Lac2A with dsLac2B (Fig 8A in 10.1073/pnas.0504982102) despite exon-specific design of dsRNA for each isoform.

Minor comments

Line 188: “In the case” seems redundant

Line 301: omit “for”

Line 317: “Lysosome” spelt incorrectly

Line 352: “but with the value 3,1 not in the female abdominal sample”

Line 520: “placed” instead of “places”

Line 522: “added” instead of “put”

Page 28: Fig 2 has “Abdominal” consistently misspelt

Rewording the title of Figure 3 will be more informative for the reader

Figure 5: Changing the color scheme to complementary colors like azure and orange, or magenta and green will provide better contrast.

- I believe the genes are incorrectly referred to as TC0101490 and TC0101489 in the Figure legend. There are some inconsistencies in gene naming and numbering, and I suggest a review of all such cases.

- It will be helpful for the reader if the authors can use dotted lines to demarcate the boundaries of individual cells in panel E. Could the authors also provide an explanation for the fluorescence observed in cell type 2 panel, for Lac2AB RNAi?

There are many other typos throughout the manuscript, but I did not note them all.

**Have all data underlying the figures and results presented in the manuscript been provided?**

Reviewer #1: Yes

Reviewer #2: Yes

Reviewer #3: Yes

PLOS authors have the option to publish the peer review history of their article (what does this mean? ). If published, this will include your full peer review and any attached files.

**Do you want your identity to be public for this peer review?** For information about this choice, including consent withdrawal, please see our Privacy Policy .

Reviewer #1: **Yes: ** Robin Beaven

Reviewer #2: No

Reviewer #3: No

**Figure resubmission:**
---

## [Decision Letter · Decision Letter 1]

21 Nov 2025

Dear Dr Wimmer,

We are pleased to inform you that your manuscript entitled "Genome-wide identification of genes involved in beetle odoriferous defensive stink gland function recognizes Laccase2 as the phenoloxidase responsible for toxic para-benzoquinone synthesis" has been editorially accepted for publication in PLOS Genetics. Congratulations!

Yours sincerely,

Giovanni Bosco, Ph.D.

Section Editor

PLOS Genetics

Giovanni Bosco

Section Editor

PLOS Genetics

Aimée Dudley

Editor-in-Chief

PLOS Genetics

Anne Goriely

Editor-in-Chief

PLOS Genetics

BlueSky: @plos.bsky.social

Comments from the reviewers (if applicable):

Reviewer's Responses to Questions

**Comments to the Authors:**

Reviewer #1: This is a detailed study revealing interesting and important insights into the function of insect defensive glands. I believe that the authors have strengthened the paper through the revisions. They have improved the clarity of the message and given a clearer sense of how their findings relate to previous studies in other insects, and the evolution of these defensive mechanisms. I did not see further changes that need to be made before publication, however I have a comment for Figure 5A which the authors may wish to modify for clarity. Their understanding of the laccase2 gene structure comes across clearly in the text but I'm not sure 5A reflects their understanding (i.e. I understand they consider TC010490 to be part of a single differentially spliced Lac2 gene). For the figure it may be clearer to show the gene structure as from the NCBI track in iBeetle-Base or similar. The figure could also show how TC010490 and TC010489 map to this, although perhaps these details are clear from the current text.

Reviewer #2: The authors have suitably improved the manuscript in this revision.

Reviewer #3: Nicely revised!

**Have all data underlying the figures and results presented in the manuscript been provided?**

Reviewer #1: Yes

Reviewer #2: Yes

Reviewer #3: None

PLOS authors have the option to publish the peer review history of their article (what does this mean? ). If published, this will include your full peer review and any attached files.

**Do you want your identity to be public for this peer review?** For information about this choice, including consent withdrawal, please see our Privacy Policy .

Reviewer #1: **Yes: ** Robin Beaven

Reviewer #2: No

Reviewer #3: No

**Data Deposition**

http://datadryad.org/submit?journalID=pgenetics&manu=PGENETICS-D-25-00083R1

**Press Queries**

---

## [Editor Report · Acceptance letter]

PGENETICS-D-25-00083R1

Genome-wide identification of genes involved in beetle odoriferous defensive stink gland function recognizes Laccase2 as the phenoloxidase responsible for toxic para-benzoquinone synthesis

Dear Dr Wimmer,

We are pleased to inform you that your manuscript entitled " 

Genome-wide identification of genes involved in beetle odoriferous defensive stink gland function recognizes Laccase2 as the phenoloxidase responsible for toxic para-benzoquinone synthesis" has been formally accepted for publication in PLOS Genetics! Your manuscript is now with our production department and you will be notified of the publication date in due course.

With kind regards,

Anita Estes

PLOS Genetics

On behalf of:
